

# Selection of housekeeping genes as internal controls for quantitative RT-PCR analysis of the veined rapa whelk (*Rapana venosa*)

Hao Song[1,2], Xin Dang[3], Yuan-qiu He[3], Tao Zhang[1,4] and Hai-yan Wang[1,4]

[1] CAS Key Laboratory of Marine Ecology and Environmental Sciences, Institute of Oceanology, Chinese Academy of Sciences, Qingdao, China
[2] University of Chinese Academy of Sciences, Beijing, China
[3] College of Fisheries, Ocean University of China, Qingdao, China
[4] Laboratory for Marine Ecology and Environmental Science, Qingdao National Laboratory for Marine Science and Technology, Qingdao, China

Corresponding authors
Tao Zhang, tzhang@qdio.ac.cn
Hai-yan Wang,
haiyanwang@qdio.ac.cn

## ABSTRACT

**Background**. The veined rapa whelk *Rapana venosa* is an important commercial shellfish in China and quantitative real-time PCR (qRT-PCR) has become the standard method to study gene expression in *R. venosa*. For accurate and reliable gene expression results, qRT-PCR assays require housekeeping genes as internal controls, which display highly uniform expression in different tissues or stages of development. However, to date no studies have validated housekeeping genes in *R. venosa* for use as internal controls for qRT-PCR.

**Methods**. In this study, we selected the following 13 candidate genes for suitability as internal controls: elongation factor-1α (*EF-1α*), α-actin (*ACT*), cytochrome c oxidase subunit 1 (*COX1*), nicotinamide adenine dinucleotide dehydrogenase (ubiquinone) 1α subcomplex subunit 7 (*NDUFA7*), 60S ribosomal protein L5 (*RL5*), 60S ribosomal protein L28 (*RL28*), glyceraldehyde 3-phosphate dehydrogenase (*GAPDH*), β-tubulin (*TUBB*), 40S ribosomal protein S25 (*RS25*), 40S ribosomal protein S8 (*RS8*), ubiquitin-conjugating enzyme E2 (*UBE2*), histone H3 (*HH3*), and peptidyl-prolyl cis-trans isomerase A (*PPIA*). We measured the expression levels of these 13 candidate internal controls in eight different tissues and twelve larvae developmental stages by qRT-PCR. Further analysis of the expression stability of the tested genes was performed using GeNorm and RefFinder algorithms.

**Results**. Of the 13 candidate genes tested, we found that *EF-1α* was the most stable internal control gene in almost all adult tissue samples investigated with *RL5* and *RL28* as secondary choices. For the normalization of a single specific tissue, we suggested that *EF-1α* and *NDUFA7* are the best combination in gonad, as well as *COX1* and *RL28* for intestine, *EF-1α* and *RL5* for kidney, *EF-1α* and *COX1* for gill, *EF-1α* and *RL28* for Leiblein and mantle, *EF-1α*, *RL5*, and *NDUFA7* for liver, *GAPDH*, *PPIA*, and *RL28* for hemocyte. From a developmental perspective, we found that *RL28* was the most stable gene in all developmental stages measured, and *COX1* and *RL5* were appropriate secondary choices. For the specific developmental stage, we recommended the following combination for normalization, *PPIA*, *RS25*, and *RL28* for stage 1, *RL5* and *RL28* for stage 2 and 5, *RL28* and *NDUFA7* for stage 3, and *PPIA* and *TUBB* for stage 4.

**Discussion**. Our results are instrumental for the selection of appropriately validated housekeeping genes for use as internal controls for gene expression studies in adult tissues or larval development of *R. venosa* in the future.

# INTRODUCTION

Gene expression analysis has great utility in increasing our understanding of gene function that underlies all biological and developmental processes. Presently available approaches or methods to evaluate gene expression include RNA *in situ* hybridization, northern blotting, microarray analysis, transcriptome sequencing, and quantitative real-time PCR (qRT-PCR). *Heid et al. (1996)* first proposed using qRT-PCR as a novel quantitative method to detect transcript levels of genes. Recently, qRT-PCR has become a common method to analyze gene expression on account of its excellent sensitivity, specificity, reproducibility, and extensive dynamic range (*Bustin et al., 2005*; *Kubista et al., 2006*).

Despite its advantages, the quality of data obtained from using this approach is dependent on RNA quality, the efficiency of reverse transcription, and appropriate normalization (*Bustin et al., 2009*). Therefore, relative qRT-PCR assay necessitates internal reference controls, which are mostly housekeeping genes. Housekeeping genes are constitutive genes that express proteins necessary to maintain elementary cellular functions. Because they have no organ or tissue specificity and are not affected in pathophysiological conditions, housekeeping genes should exhibit stable expression levels under various experimental conditions and in different tissues and developmental stages (*Butte, Dzau & Glueck, 2002*; *Eisenberg & Levanon, 2003*). Several housekeeping genes with relatively constant expression are considered as internal controls in qRT-PCR. These include glyceraldehyde 3-phosphate dehydrogenase (*GAPDH*), ribosomal protein (*RP*), tubulin (*TUB*), actin (*ACT*), elongation factor (*EF*), ubiquitin (*UBQ*), and histone H3 (*HH3*) (*Bangaru et al., 2012*; *Huggett et al., 2005*; *Lee et al., 2010*; *Ray & Johnson, 2014*; *Wang et al., 2012*). However, various studies have reported that internal standards, mainly housekeeping genes used in quantifying mRNA expression, exhibit variable expression levels under different tissue types, developmental stages, and environmental conditions (*Stürzenbaum & Kille, 2001*; *Thellin et al., 1999*). Because selection of the appropriate internal control relies on the type of samples measured in the experiment, it is necessary to identify and characterize housekeeping genes which are essential for qRT-PCR data normalization in the experiment in question.

The veined rapa whelk *Rapana venosa*, which is an economically important mollusk in China, has been bred since 1992 (*Yuan, 1992*). In France, Argentina, and the United States, *R. venosa* is considered an invasive pest that severely disrupts the survival of native bivalves because of its lack of human consumption (*Culha et al., 2009*; *Giberto et al., 2006*; *Leppäkoski, Gollasch & Olenin, 2002*; *Mann & Harding, 2003*; *Mann, Harding & Westcott, 2006*). Because of its commercial importance and ecological impact, molecular research in

the morphology and biology of *R. venosa* have been increasing, and qRT-PCR is commonly being used as the tool to study gene expression (*Lu et al., 2008*; *Samadi & Steiner, 2009*). Housekeeping genes need to be identified and validated as reliable reference genes, however, no such prior study has been carried out in *R. venosa*. Thus, the objective of this reference-selection study is to evaluate 13 candidate reference genes associated with eight target tissues and 12 developmental stages in *R. venosa*.

## MATERIALS AND METHODS

### Larvae culture and sample collection

Egg capsules of *R. venosa* were collected naturally from Laizhou Bay, Laizhou, China. Following published methods, larvae were incubated in appropriately sized tanks at Blue Ocean Co. Limited (Laizhou, China) (*Pan et al., 2013*). Newly hatched pelagic larvae were transferred to 2.5 m × 2.5 m × 1.5 m tanks with a density range of 0.3–0.05 ind/mL, determined by developmental stage. Larvae were fed a mixture of microalgae containing *Platymonas subcordiformis*, *Isochrysis galbana*, and *Chlorella vulgaris* ($13.0 \times 10^4$ cells/mL daily). Seawater was treated by sand filtration and UV irradiation before samples were cultured. Seawater temperature was below $25 \pm 1\,°C$. Larvae samples were examined by microscope to ensure synchronous growth in developmental stages including blastula, juvenile, and adult stages. Samples were collected and washed with distilled water, frozen in liquid nitrogen, and stored at $-80\,°C$ until use. We selected five biological replicates from 12 larval stage ((blastula (L), gastrula (M), trochophore (N), early intra-membrane veliger (R), mid intra-membrane veliger (S), late intra-membrane veliger (T), one-spiral whorl larvae (C), two-spiral larvae (D), early three-spiral whorl larvae (F), late three-spiral whorl larvae (G), four-spiral whorl larvae (J), and juvenile stage (Y)) and all tested tissues (gill, hemocyte, intestine, Leiblein's gland, liver, kidney, mantle, and gonad) were aseptically dissected from five adult specimens. Hemolymph was extracted from the pericardial cavity using a 1 mL medical injector. Hemocytes were obtained by centrifugation at $4\,°C$ and $1,000\times$ g for 10 min.

### Total RNA extraction and cDNA synthesis

Total RNA was extracted from tissue samples of gills, intestines, Leiblein's glands, livers, kidneys, and gonads, and from larvae of different developmental stages using MiniBEST Universal RNA Extraction Kit (TaKaRa, Tokyo, Japan), and from hemocyte and mantle using RNAiso Plus (TaKaRa) according to manufacturer's instructions. RNA integrity was confirmed by gel electrophoresis based on the predicted product size. RNA from each sample was diluted with nuclease-free water and 0.1 µg RNA of each sample was used as the template for cDNA synthesis using a PrimeScript$^{TM}$ RT reagent Kit with gDNA Eraser (TaKaRa). Prior to qRT-PCR, cDNA was diluted 10-fold.

### Selection of candidate internal controls

According to RNA-seq transcriptome data of developmental samples, which were derived from stages C to Y and performed in triplicate for each stage (*Song et al., 2016*), genes with similar expression patterns were identified and classified to different clusters. Candidate housekeeping genes were selected from clusters exhibiting expression stability based the

RPKM value in different developmental stages. In total, 13 genes were selected using a previously published RNA-seq data (*Song et al., 2016*).

## Primer design and qRT-PCR

The primers for qRT-PCR were designed using Primer Premier 5 (PREMIER Biosoft, USA) and are listed in Table 1. qRT-PCR was performed using a SYBR Green® real-time PCR assay consisting of a SYBRPrimeScript$^{TM}$ RT-PCR Kit II (TaKaRa) with a Mastercycler® ep realplex S (Eppendorf; Hamburg, Germany). Amplifications were carried out in a total volume of 20 µL (10 µL of SYBR Green Master Mix, 0.4 µL of each forward and reverse primer (10 µmol/L), 1 µL of diluted cDNA, and 8.2 µL RNase-free water) as follows: 95 °C for 2 min followed by 40 cycles of 95 ° C for 15 s, the respective annealing temperature (Tm) for 15 s, and 68 °C for 20 s. The Tm for GAPDH, EF-1α, ACT, COX1, NDUFA7, and RL5 is 58.5 °C, the Tm for RL28, TUBB, RS25, RS8, UBE2, and PPIA is 57.5 °C, and the Tm for HH3 is 56.5 °C. Melting-curve analysis of the amplification products was performed and following electrophoresis each gel picture was analyzed to confirm the product by the predicted size. Each assay was performed in triplicate and *Cq* values were recorded for further analysis.

## Analysis of gene expression stability

The expression stability of 13 candidate housekeeping genes among the different RNA samples was calculated using the Excel-based tool GeNorm v3.4 (https://genorm.cmgg.be/) and overall stability of these candidate genes was determined using RefFinder (http://fulxie.0fees.us/?type=reference&i=1). We used this suite of tools to ensure a statistically thorough analysis and robust identification of housekeeping genes for use in qRT-PCR of *R. venosa*.

GeNorm evaluates gene stability ($M$) of inputted genes using a statistical algorithm according to geometric averaging of multiple control genes and means pairwise variation of a gene from those remaining control genes in all provided samples. *Vandesompele et al. (2002)* proposed a value of 1.5 as a cut-off for suitability as an endogenous control, especially with heterogeneous samples such as different cell types or tissues. Based on this approach, genes with the lowest $M$ value have the highest expression stability. The best combinations of two internal control genes with a constant level were selected by stepwise exclusion of the gene with the highest $M$ value followed by a recalculation of new $M$ values for all of the remaining genes (*Vandesompele et al., 2002*). In addition, GeNorm is used to determine the optimal number of housekeeping genes by pairwise number variation analysis. It computes the geometric mean of the selected genes that are expressed steadily for accurate normalization. A pairwise variation below 0.15, which is determined by $Vn/n+1$, means that an added control gene ($n+1$) would not further improve the normalization factor (*Vandesompele et al., 2002*).

RefFinder is an online analysis tool, which includes GeNorm v3.4, NormFinder v20, BestKeeper v1, and delta CT, that is used to avoid one-sidedness and the potential limitations of relying on a single tool or algorithm for stability analysis to identify reference genes of interest (*Pfaffl et al., 2004*). *Andersen, Jensen & Ørntoft (2004)* developed NormFinder

Song et al. (2017), *PeerJ*, DOI 10.7717/peerj.3398

**Table 1** Candidate housekeeping genes and their primer sequences for qRT-PCR.

| Accession | Gene name | Gene symbol | Biological function | Primer sequence (5′–3′) | Tm (°C) | Product size (bp) | Efficiency (%) |
|---|---|---|---|---|---|---|---|
| c85865_g1 | Elongation factor-1α | EF-1α | Essential component of the eukaryotic translational apparatus | f: CGAGATCAAGGAGAAGTGCG<br>r: CAACGGTCTGCTTCATGTCA | 58.5 | 182 | 101.4 |
| c154556_g1 | α-actin | ACT | Cytoskeletal structural protein | f: CGAGAACAGGTACACGCAAT<br>r: GTTGAAGGACATGCGGAACT | 58.5 | 166 | 97.5 |
| c104226_g1 | Cytochrome c oxidase subunit 1 | COX1 | Respiratory electron transport chain of mitochondria | f: CTCCTGATATAGCTTTCCCTCG<br>r: CTACAGAACCACCAGCATGAG | 58.5 | 165 | 97.7 |
| c205682_g1 | Nicotinamide adenine dinucleotide dehydrogenase [ubiquinone] 1α subcomplex subunit 7 | NDUFA7 | Electron transport in the respiratory chain | f: GGGAGACGGGAAAACTTGAC<br>r: AGGGAGGTGACAATTAGCCA | 58.5 | 152 | 99.8 |
| c121877_g2 | 60S ribosomal protein L5 | RL5 | 60S ribosomal subunit | f: GTGGAGGATGAGGATGGACA<br>r: CTGCCTTGTACTCGTTGGTC | 58.5 | 184 | 96.0 |
| c64332_g1 | 60S ribosomal protein L28 | RL28 | 60S ribosomal subunit | f: CGTGCGTAACATCACCAAGA<br>r: CACCACAGCTACCACACATT | 57.5 | 151 | 102.4 |
| c149072_g1 | Glyceraldehyde 3-phosphate dehydrogenase | GAPDH | Glycolytic enzyme | f: CTCTACCAGTCAACGCTCCA<br>r: AATGCGACACCCATCAGAGA | 58.5 | 138 | 97.6 |
| c150134_g1 | β-tubulin | TUBB | Structural protein | f: CACTTTCGTGGGCAACTC<br>r: ACTCGGACACCAGGTCGT | 57.5 | 170 | 96.0 |
| c64251_g1 | 40S ribosomal protein S25 | RS25 | 40S ribosomal subunit | f: ACAAGATGCTGAAGGAGGTG<br>r: ACGCCAGACAAACATGAAAA | 57.5 | 224 | 99.5 |
| c112836_g1 | 40S ribosomal protein S8 | RS8 | 40S ribosomal subunit | f: TGGTGAAGTCCTGCATCG<br>r: CCTGGGCTGACAGTTTGA | 57.5 | 111 | 97.6 |
| c68556_g1 | Ubiquitin-conjugating enzyme E2 | UBE2 | Protein degradation | f: TTCCTGGACAACTGTCATTCT<br>r: CCTTTCCTCCCTACATTATCTT | 57.5 | 378 | 100.3 |
| c105544_g1 | Histone H3 | HH3 | Essential structure of nucleosome | f: TACAGGCAGCAGATCAGGTT<br>r: CCAGGAAAGTAAGGAGCAGAG | 56.5 | 366 | 99.0 |
| c45955_g1 | Peptidyl prolyl cis-trans isomerase A | PPIA | Immunoregulation | f: ATCAGAACACGCTTTCCTTT<br>r: CCAAACCCTTTCTCACCAG | 57.5 | 235 | 97.8 |
to estimate both overall and subgroup variation of the sample set of candidate genes for normalization factors (NFs) in a gene expression study. Three candidate housekeeping genes and two tested samples per group are required to set the minimum input data. Raw *Cq* values are first log-transformed and used as input. Random co-regulated genes would not bias the results of the software. NormFinder ranks the best candidate reference genes according to the lowest expression stability values and the lowest variation values by combining intra- and inter-group variability.

BestKeeper calculates descriptive statistics of the *Cq* values and Pearson correlation coefficient (*Pfaffl et al., 2004*). Internal control genes with stable expression for use as a housekeeping gene are identified based on highly correlated expression levels. The correlation between each candidate reference gene and the BestKeeper index, which is determined by calculating the geometric mean of the *Cq* values of the candidate genes, is estimated by the Pearson correlation coefficient ($r$), the coefficient of determination ($r^2$), and *P* value. BestKeeper ranks the candidate housekeeping genes according to Cq variation, which is displayed as standard deviation (SD), $r$, and as $r^2$ with the BestKeeper index value. An SD threshold value less than 1.0 is recommended by *Pfaffl et al. (2004)* and the closer $r^2$ is to 1, the better.

## RESULTS

### Selection of housekeeping genes to be used as internal controls

Eight different subclusters that exhibited various gene expression patterns were identified. Genes that have similar gene expression patterns were evaluated based on their correlation and classified as a single subcluster (Fig. 1). We found that each subcluster had between 179 and 8,222 genes. The gene expression patterns of subclusters 1 and 8 were highly variable in six developmental stages, whereas genes in subclusters 2, 3, and 4 indicated the greatest expression stability. Therefore, genes from these latter subclusters were selected as eligible candidate housekeeping genes. TUBB, which is a common housekeeping gene already used as an internal control, was selected from subcluster 2. ACT, UBE2, and GAPDH were selected from subcluster 3, while the other nine candidate housekeeping genes were from subcluster 4.

### Real-time PCR amplification of housekeeping genes

Single peaks of the melting curves in different samples confirmed primer accuracy and gene-specific amplification (Fig. S1). In addition, agarose gel electrophoresis exhibited a single band for each amplified gene and PCR product were confirmed based on the expected size.

### Gene expression stability analysis in tissues

In our study, we identified 13 candidate reference genes and tested them for expression stability in eight different tissues, specifically, gill, hemocyte, intestine, Leiblein's gland (Leiblein), liver, kidney, mantle, and gonad from five adult individuals.

We analyzed the raw quantification cycle data (*Cq* values) obtained from qRT-PCR and the determined variation among these candidate housekeeping genes in trial samples. In all tissues, we found that the average *Cq* values of the 13 genes ranged from 14.71 to

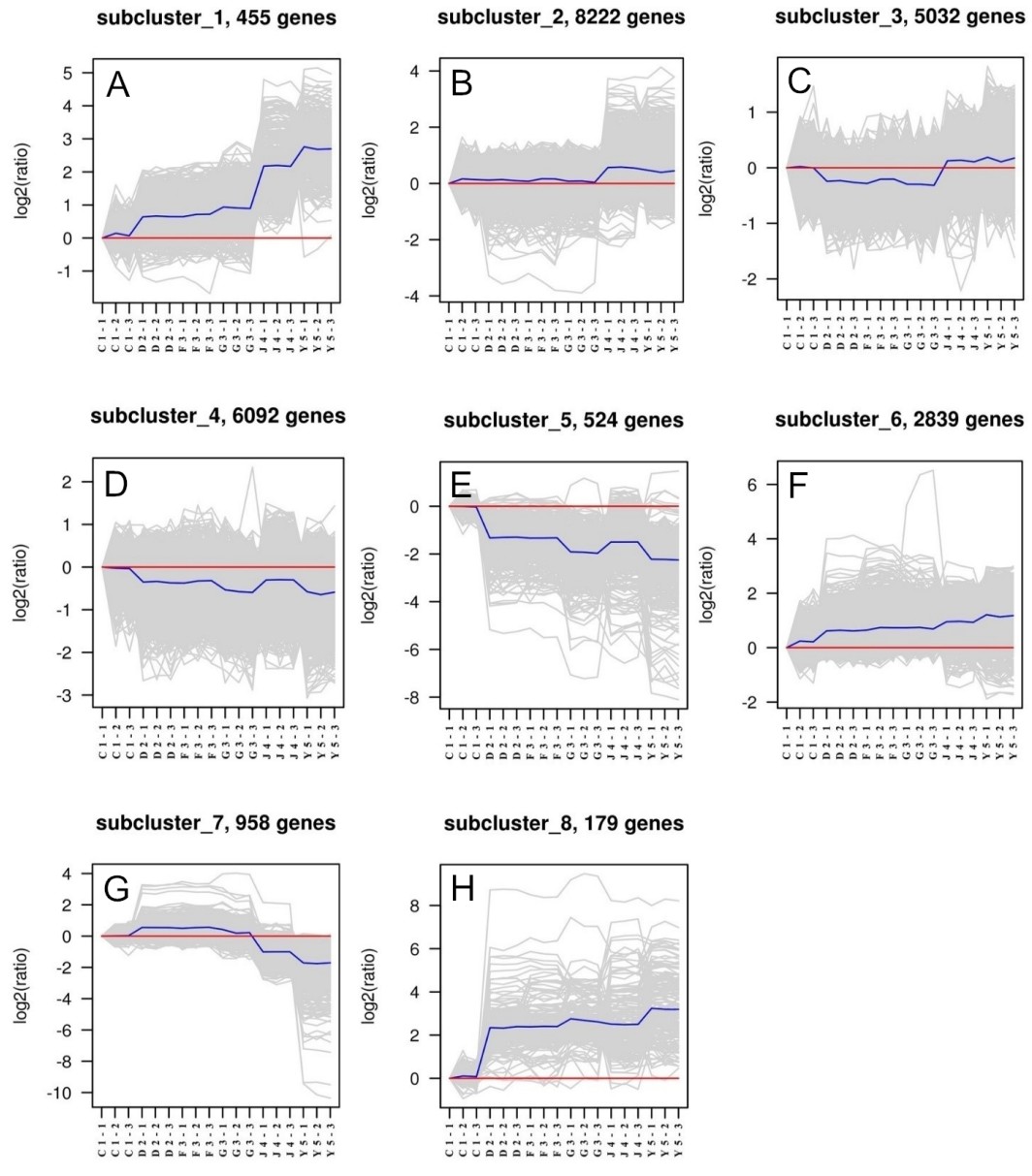

**Figure 1  Clusters of genes in different developmental samples.** Six developmental stages in triplicates consisting of one-spiral whorl stage (C), two-spiral whorl stage (D), early three-spiral whorl stage (F), late three-spiral whorl stage (G), four-spiral whorl stage (J), and juveniles (Y).

33.45 (Table 2). We found that *COX1* had the lowest mean *Cq* values, which represent the highest expression levels, both in all tissues and in individual tissues, whereas HH3 had the highest mean *Cq* values, which indicates the lowest expression levels, in various tissues with the exception of the mantle. We found that each housekeeping gene displayed minor variability in its expression level in the various tissues under the same conditions. According to computed values of standard error (SE), we found that *RL28* (SE = 0.37), *EF-1α* (SE = 0.38), *NDUFA7* (SE = 0.39), and *RS25* (SE = 0.39) have the least varying transcript abundance values when all of the tissues were analyzed together; however, we found that

Song et al. (2017), *PeerJ*, DOI 10.7717/peerj.3398

**Table 2  Tissue-specific expression profiles of candidate reference genes.** Data are shown as raw *Cq* values and represented as mean ± standard error (SE).

| Gene | All tissues | Gill | Hemocyte | Intestine | Leiblein | Liver | Kidney | Mantle | Gonad |
|------|-------------|------|----------|-----------|----------|-------|--------|--------|-------|
| *EF-1α* | 20.70 ± 0.38 | 19.14 ± 0.36 | 21.81 ± 0.74 | 23.00 ± 1.35 | 22.48 ± 0.92 | 20.35 ± 0.46 | 20.73 ± 0.68 | 16.98 ± 0.61 | 21.09 ± 0.46 |
| *ACT* | 26.50 ± 0.47 | 25.20 ± 0.45 | 28.51 ± 0.77 | 29.24 ± 1.35 | 28.24 ± 0.85 | 26.88 ± 0.52 | 26.56 ± 0.73 | 20.57 ± 0.44 | 26.80 ± 0.55 |
| *COX1* | 19.26 ± 0.41 | 17.36 ± 0.31 | 20.39 ± 0.97 | 21.47 ± 1.25 | 21.49 ± 0.91 | 18.57 ± 0.56 | 19.10 ± 0.77 | 15.36 ± 0.48 | 20.32 ± 0.68 |
| *NDUFA7* | 26.92 ± 0.39 | 24.69 ± 0.32 | 28.42 ± 0.79 | 29.49 ± 1.19 | 28.48 ± 1.00 | 26.54 ± 0.56 | 27.40 ± 0.60 | 23.41 ± 0.73 | 26.94 ± 0.47 |
| *RL5* | 23.14 ± 0.41 | 21.92 ± 0.42 | 24.19 ± 0.81 | 25.91 ± 1.45 | 24.57 ± 0.95 | 22.83 ± 0.44 | 23.89 ± 0.67 | 18.74 ± 0.58 | 23.05 ± 0.53 |
| *RL28* | 22.59 ± 0.37 | 21.47 ± 0.41 | 23.65 ± 0.52 | 24.96 ± 1.19 | 24.18 ± 0.87 | 22.13 ± 0.42 | 23.42 ± 0.75 | 18.63 ± 0.58 | 22.31 ± 0.61 |
| *GAPDH* | 24.14 ± 0.42 | 21.60 ± 0.44 | 24.18 ± 0.71 | 26.95 ± 1.11 | 24.98 ± 0.95 | 24.19 ± 0.54 | 24.72 ± 0.59 | 20.11 ± 0.63 | 26.44 ± 0.84 |
| *TUBB* | 25.95 ± 0.43 | 23.16 ± 0.58 | 26.72 ± 0.35 | 25.78 ± 1.37 | 24.95 ± 0.72 | 23.42 ± 0.78 | 26.08 ± 0.80 | 28.00 ± 0.71 | 29.48 ± 1.33 |
| *PPIA* | 25.56 ± 0.41 | 24.43 ± 0.35 | 26.15 ± 0.57 | 28.41 ± 1.19 | 26.74 ± 1.03 | 24.78 ± 0.72 | 25.32 ± 0.56 | 21.67 ± 0.71 | 26.99 ± 1.20 |
| *RS25* | 24.95 ± 0.39 | 24.27 ± 0.26 | 24.95 ± 0.42 | 27.18 ± 1.39 | 26.22 ± 0.91 | 24.42 ± 0.62 | 25.70 ± 0.80 | 20.59 ± 0.45 | 26.30 ± 0.82 |
| *UBE2* | 26.75 ± 0.48 | 24.49 ± 0.46 | 26.60 ± 0.79 | 29.71 ± 1.46 | 28.58 ± 0.85 | 26.65 ± 0.57 | 27.19 ± 1.11 | 22.09 ± 0.88 | 28.67 ± 1.18 |
| *RS8* | 24.15 ± 0.43 | 22.95 ± 0.61 | 24.45 ± 0.77 | 26.74 ± 1.52 | 25.80 ± 0.79 | 23.50 ± 0.57 | 24.57 ± 0.85 | 19.81 ± 0.55 | 25.42 ± 0.97 |
| *HH3* | 30.36 ± 0.41 | 28.45 ± 0.42 | 30.13 ± 0.69 | 32.84 ± 0.44 | 32.25 ± 0.80 | 30.58 ± 0.53 | 30.48 ± 0.78 | 26.04 ± 1.05 | 32.14 ± 0.93 |

different housekeeping genes displayed variable levels of transcript abundance in different tissues. The genes with the lowest SE in the eight tissues examined was *RS25* (SE = 0.26) in gill, *TUBB* (SE = 0.35 and SE = 0.72) in hemocyte and Leiblein respectively, *HH3* (SE = 0.44) in intestine, *RL5* (SE = 0.44) and *RL28* (SE = 0.42) in liver, *PPIA* (SE = 0.56) in kidney, *ACT* (SE = 0.44) and *RS25* (SE = 0.45) in mantle, and *EF-1α* (SE = 0.46) and *NDUFA7* (SE = 0.47) in gonad. These findings demonstrate that no single candidate housekeeping gene is expressed at a stable level on the basis of *Cq* value only in these eight tissues from different adult *R. venosa* samples, and therefore, it is necessary to select better-suited housekeeping genes using additional statistical analyses.

The GeNorm-derived *M* values of candidate housekeeping genes in all tissues and for each tissue are shown in Fig. 2. We found that *EF-1α* and *RL5* (both *M* = 0.52) showed the highest stability in all tissues as well as in gonads, kidneys, and livers. However, we found that *EF-1α* and *RL28* were the best combination of two internal control genes for gill, intestine, and mantle, whereas the best control gene pairs for hemocyte and Leiblein were *PPIA* and *HH3*, and *RL28* and *UBE2*, respectively. These results indicate that *EF-1α*, *RL5*, and *RL28* are the most appropriate internal control genes for most tissues.

Figure 2 also shows a ranking of the stability values calculated by NormFinder for tissue-specific housekeeping genes. The best housekeeping genes on the basis of all tissues together was *EF-1α* with a stability value of 0.63 as well as in Leiblein and mantle (0.16 and 0.10, respectively), whereas in the gill and intestine it was *COX1* (0.08 and 0.20, respectively), in hemocyte and gonad it was *RL28* (0.68 and 0.84, respectively), and in liver and kidney it was both *RL5* and *EF-1α* (both 0.17 and 0.05, respectively). Based on these findings, *EF-1α* is an appropriate stable internal control gene for all tissues analyzed either together or separately. In addition, *RL5*, *COX1*, and *RL28* can be used for their respective tissue-specific analyses.

Using SD values generated from BestKeeper for each of the various housekeeping genes, we found major differences in each tissue (shown in Tables 3 and 4). *RL28* was identified as the best gene for all tissues together and in livers, whereas *RS25* was identified for use in gills and mantles, *TUBB* in hemocyte and Leiblein, *HH3* in intestines, *PPIA* in kidneys, and *NDUFA7* in gonads. However, as ranked by *r*, we found that *EF-1α* was the most stably expressed gene in all tissues together and in most separate tissues, namely, gills, Leiblein, mantle, gonads, and kidneys, although *EF-1α* and *RL5* had the same rank position in kidneys. In addition, *RL5* was the best gene in hemocyte, whereas *COX1* and *PPIA* were ideal for intestines and livers, respectively. Therefore, based on our findings with *r*, *EF-1α* was the most stable housekeeping gene in most tissue samples.

## Gene expression stability analysis in developmental stages

In this study, using five specimens, we identified 13 candidate reference genes and tested them for expression stability in 12 different developmental larval stages. For data description and display in tables and figures, we merged these 12 developmental larval stages into five groups/stages according to their developmental characteristics: stage 1 (L, M, N); stage 2 (R, S, T); stage 3 (C, D, F); stage 4 (G, J); and stage 5 (Y).

We analyzed the *Cq* values obtained from qRT-PCR and calculated variation among the candidate housekeeping genes evaluated in the samples. In all stages combined, the
**Table 3 Ranking of candidate reference genes in order of expression stability calculated by BestKeeper for different tissues.** Data shown as Pearson correlation coefficient (*r*) and standard deviation (SD).

| Rank | All tissues | | Gill | | Hemocyte | | Intestine | | Leiblein | | Liver | | Kidney | | Mantle | | Gonad | |
|---|---|---|---|---|---|---|---|---|---|---|---|---|---|---|---|---|---|---|
| | SD | r | SD | r | SD | r | SD | r | SD | r | SD | r | SD | r | SD | r | SD | r |
| 1 | RL28 | EF-1α | RS25 | EF-1α | TUBB | RL5 | HH3 | COX1 | TUBB | EF-1α | RL28 | PPIA | PPIA | EF-1α | RS25 | EF-1α | NDUFA7 | EF-1α |
| 2 | EF-1α | RS8 | COX1 | RS8 | RS25 | RL28 | GAPDH | TUBB | RS8 | RL28 | GAPDH | RL5 | GAPDH | RL5 | COX1 | RL5 | EF-1α | RS8 |
| 3 | RS25 | RL28 | NDUFA7 | RL28 | RL28 | GAPDH | NDUFA7 | RL5 | COX1 | UBE2 | RL5 | NDUFA7 | NDUFA7 | UBE2 | ACT | COX1 | RL5 | RL28 |
| 4 | RL5 | RL5 | EF-1α | RL5 | PPIA | EF-1α | RL28 | EF-1α | ACT | RL5 | EF-1α | EF-1α | RL5 | PPIA | RL28 | PPIA | ACT | RL5 |
| 5 | NDUFA7 | COX1 | PPIA | COX1 | RS8 | ACT | PPIA | RL28 | UBE2 | GAPDH | HH3 | COX1 | RL28 | NDUFA7 | RL5 | RL28 | COX1 | COX1 |
| 6 | PPIA | UBE2 | ACT | PPIA | EF-1α | NDUFA7 | COX1 | PPIA | RL28 | TUBB | UBE2 | RS8 | EF-1α | ACT | EF-1α | UBE2 | RL28 | PPIA |
| 7 | HH3 | PPIA | HH3 | ACT | ACT | RS8 | ACT | RS25 | RS25 | RS8 | RS8 | RL28 | ACT | RL28 | RS8 | NDUFA7 | RS25 | ACT |
| 8 | RS8 | RS25 | RL28 | HH3 | GAPDH | COX1 | EF-1α | UBE2 | NDUFA7 | NDUFA7 | ACT | RS25 | HH3 | RS25 | GAPDH | RS8 | HH3 | HH3 |
| 9 | GAPDH | NDUFA7 | RL5 | UBE2 | HH3 | PPIA | RS25 | ACT | GAPDH | RS25 | NDUFA7 | ACT | TUBB | COX1 | PPIA | HH3 | GAPDH | UBE2 |
| 10 | COX1 | GAPDH | GAPDH | NDUFA7 | RL5 | HH3 | TUBB | GAPDH | EF-1α | ACT | COX1 | TUBB | RS25 | GAPDH | TUBB | GAPDH | RS8 | NDUFA7 |
| 11 | TUBB | ACT | UBE2 | RS25 | NDUFA7 | UBE2 | RL5 | RS8 | HH3 | COX1 | RS25 | UBE2 | COX1 | RS8 | NDUFA7 | RS25 | UBE2 | RS25 |
| 12 | ACT | HH3 | TUBB | GAPDH | UBE2 | RS25 | UBE2 | NDUFA7 | RL5 | PPIA | TUBB | GAPDH | RS8 | TUBB | UBE2 | TUBB | PPIA | GAPDH |
| 13 | UBE2 | TUBB | RS8 | TUBB | COX1 | TUBB | RS8 | HH3 | PPIA | HH3 | PPIA | HH3 | UBE2 | HH3 | HH3 | ACT | TUBB | TUBB |

Song et al. (2017), *PeerJ*, DOI 10.7717/peerj.3398

**Table 4** **Results from BestKeeper descriptive statistical analysis and BestKeeper regression analysis in different tissues and developmental stages (correlation coefficients between each control gene *Cq* and the BestKeeper Index).** Geo Mean represents Geometric mean while AR Mean represents Arithmetic mean (*Cq*). Tissues and developmental stages are represented as T and D, respectively.

| | EF-1α | | ACT | | COX1 | | NDUFA7 | | RL5 | | RL28 | | GAPDH | | TUBB | | PPIA | | RS25 | | UBE2 | | RS8 | | HH3 | |
|---|---|---|---|---|---|---|---|---|---|---|---|---|---|---|---|---|---|---|---|---|---|---|---|---|---|---|
| | T | D | T | D | T | D | T | D | T | D | T | D | T | D | T | D | T | D | T | D | T | D | T | D | T | D |
| *n* | 40 | 60 | 40 | 60 | 40 | 60 | 40 | 60 | 40 | 60 | 40 | 60 | 40 | 60 | 40 | 60 | 40 | 60 | 40 | 60 | 40 | 60 | 40 | 60 | 40 | 60 |
| Geo mean (*Cq*) | 20.56 | 18.95 | 26.33 | 24.39 | 19.09 | 19.09 | 26.81 | 25.88 | 22.99 | 22.74 | 22.47 | 22.63 | 24.00 | 22.79 | 25.81 | 22.70 | 25.43 | 25.36 | 24.83 | 25.25 | 26.58 | 26.74 | 24.01 | 24.17 | 30.25 | 28.40 |
| AR mean (*Cq*) | 20.70 | 19.12 | 26.50 | 24.61 | 19.26 | 19.29 | 26.92 | 26.00 | 23.14 | 22.91 | 22.59 | 22.80 | 24.14 | 22.93 | 25.95 | 23.09 | 25.56 | 25.50 | 24.95 | 25.44 | 26.75 | 26.90 | 24.15 | 24.39 | 30.36 | 28.51 |
| Min (*Cq*) | 15.79 | 15.89 | 19.33 | 20.50 | 14.24 | 14.71 | 22.13 | 22.60 | 17.54 | 18.80 | 17.39 | 18.85 | 18.13 | 18.81 | 21.64 | 17.63 | 19.96 | 21.45 | 19.63 | 21.02 | 20.87 | 22.79 | 18.65 | 19.62 | 23.56 | 24.72 |
| Max (*Cq*) | 27.26 | 25.67 | 33.86 | 32.70 | 25.34 | 25.40 | 32.45 | 32.41 | 30.57 | 29.67 | 28.72 | 28.76 | 29.84 | 29.89 | 32.73 | 33.20 | 31.46 | 32.15 | 31.12 | 32.31 | 33.62 | 32.85 | 31.61 | 31.89 | 34.08 | 33.45 |
| SD (±*Cq*) | 1.88 | 2.30 | 2.24 | 2.98 | 2.11 | 2.52 | 1.93 | 2.27 | 1.90 | 2.54 | 1.79 | 2.59 | 2.09 | 2.24 | 2.17 | 3.79 | 1.99 | 2.39 | 1.88 | 2.81 | 2.39 | 2.64 | 2.07 | 2.99 | 2.06 | 2.20 |
| CV (% *Cq*) | 9.08 | 12.02 | 8.46 | 12.10 | 10.96 | 13.04 | 7.15 | 8.73 | 8.21 | 11.09 | 7.93 | 11.34 | 8.65 | 9.75 | 8.36 | 16.43 | 7.79 | 9.36 | 7.54 | 11.05 | 8.94 | 9.82 | 8.58 | 12.27 | 6.79 | 7.71 |
| Min (*x*-fold) | −27.35 | −8.37 | −127.64 | −14.85 | −28.81 | −20.86 | −25.66 | −9.71 | −43.81 | −15.38 | −33.88 | −13.78 | −58.46 | −15.73 | −18.02 | −33.51 | −44.43 | −15.02 | −36.79 | −18.81 | −52.21 | −15.50 | −41.00 | −23.48 | −103.17 | −12.86 |
| Max (*x*-fold) | 103.71 | 105.07 | 185.34 | 316.90 | 76.20 | 79.21 | 49.82 | 92.41 | 190.90 | 121.70 | 76.00 | 69.80 | 57.31 | 137.63 | 120.96 | 1451.54 | 65.19 | 110.77 | 78.17 | 133.14 | 131.93 | 68.88 | 194.32 | 210.36 | 14.23 | 33.02 |
| Std dev (±*x*-fold) | 3.68 | 4.92 | 4.73 | 7.87 | 4.32 | 5.72 | 3.80 | 4.82 | 3.73 | 5.82 | 3.46 | 6.01 | 4.25 | 4.71 | 4.50 | 13.87 | 3.98 | 5.24 | 3.69 | 7.02 | 5.24 | 6.24 | 4.20 | 7.96 | 4.17 | 4.59 |
| Coeff. of corr (*r*) | 0.96 | 0.98 | 0.92 | 0.98 | 0.95 | 0.99 | 0.93 | 0.99 | 0.95 | 0.99 | 0.95 | 1.00 | 0.93 | 0.95 | 0.30 | 0.96 | 0.94 | 0.98 | 0.94 | 0.99 | 0.95 | 0.99 | 0.96 | 0.99 | 0.87 | 0.87 |
| Coeff. of det (*r*²) | 0.93 | 0.95 | 0.84 | 0.96 | 0.90 | 0.98 | 0.87 | 0.97 | 0.90 | 0.98 | 0.91 | 0.99 | 0.87 | 0.91 | 0.09 | 0.93 | 0.88 | 0.96 | 0.88 | 0.98 | 0.90 | 0.97 | 0.92 | 0.98 | 0.75 | 0.75 |
| *P* value | 0.001 | 0.001 | 0.001 | 0.001 | 0.001 | 0.001 | 0.001 | 0.001 | 0.001 | 0.001 | 0.001 | 0.001 | 0.001 | 0.001 | 0.06 | 0.06 | 0.001 | 0.001 | 0.001 | 0.001 | 0.001 | 0.001 | 0.001 | 0.001 | 0.001 | 0.001 |

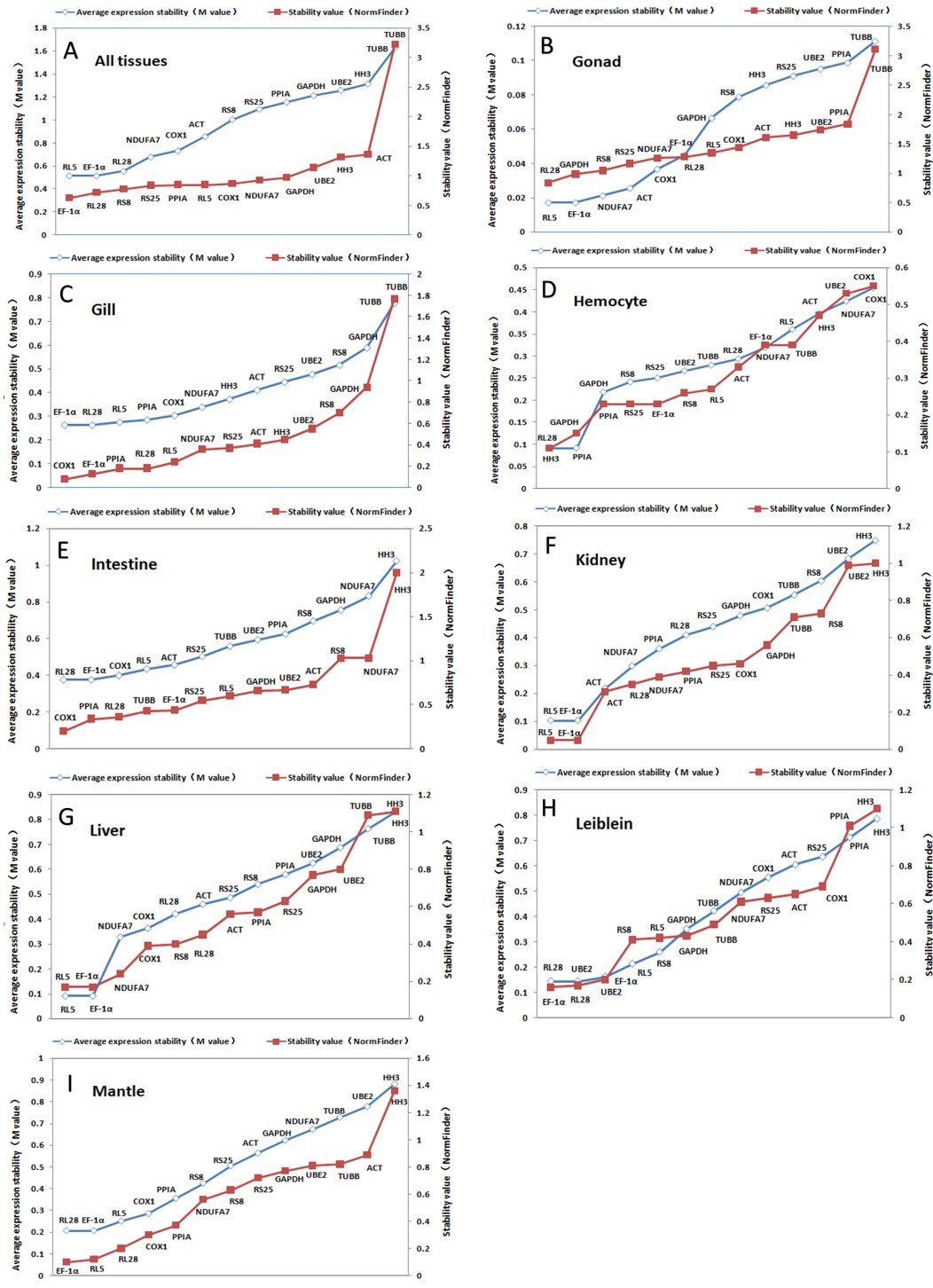

**Figure 2  Ranking of candidate housekeeping genes in adult tissue.** NormFinder (stability value, filled squares) and GeNorm (average expression stability (*M* value) of remaining genes, open rhombus) ranking of candidate housekeeping genes in eight tissues together and separately. A lower value indicates more stable expression.

**Table 5  Developmental stage-specific expression profiles of candidate reference genes.** Data are shown as raw *Cq* values and represented as mean ± standard error (SE).

| Gene | All stages | Stage 1 | Stage 2 | Stage 3 | Stage 4 | Stage 5 |
|------|-----------|---------|---------|---------|---------|---------|
| GAPDH | 22.93 ± 0.35 | 26.39 ± 0.54 | 22.83 ± 0.64 | 21.43 ± 0.22 | 21.32 ± 0.21 | 20.61 ± 0.48 |
| EF-1α | 19.12 ± 0.34 | 22.35 ± 0.39 | 19.59 ± 0.65 | 17.26 ± 0.19 | 17.05 ± 0.20 | 17.80 ± 0.51 |
| ACT | 24.61 ± 0.43 | 29.09 ± 0.44 | 25.02 ± 0.73 | 22.23 ± 0.22 | 22.38 ± 0.20 | 21.50 ± 0.51 |
| COX1 | 19.29 ± 0.36 | 22.93 ± 0.32 | 19.71 ± 0.68 | 16.87 ± 0.18 | 17.70 ± 0.17 | 17.49 ± 0.33 |
| NDUFA7 | 26.00 ± 0.34 | 29.41 ± 0.41 | 26.29 ± 0.58 | 23.90 ± 0.19 | 24.69 ± 0.11 | 23.86 ± 0.38 |
| RL5 | 22.91 ± 0.37 | 26.73 ± 0.38 | 23.10 ± 0.67 | 20.63 ± 0.20 | 21.24 ± 0.11 | 21.07 ± 0.33 |
| RL28 | 22.80 ± 0.37 | 26.67 ± 0.31 | 23.17 ± 0.64 | 20.33 ± 0.17 | 21.07 ± 0.12 | 21.02 ± 0.24 |
| TUBB | 23.09 ± 0.57 | 29.46 ± 0.66 | 22.82 ± 0.77 | 19.03 ± 0.15 | 20.09 ± 0.10 | 22.98 ± 0.28 |
| RS25 | 25.44 ± 0.41 | 29.90 ± 0.37 | 25.48 ± 0.69 | 22.92 ± 0.15 | 23.52 ± 0.08 | 23.32 ± 0.19 |
| RS8 | 24.39 ± 0.44 | 29.02 ± 0.45 | 24.55 ± 0.75 | 21.48 ± 0.17 | 22.76 ± 0.11 | 22.08 ± 0.37 |
| UBE2 | 26.90 ± 0.38 | 30.83 ± 0.37 | 27.35 ± 0.66 | 24.35 ± 0.17 | 25.47 ± 0.11 | 24.22 ± 0.25 |
| PPIA | 25.50 ± 0.36 | 29.27 ± 0.38 | 25.51 ± 0.62 | 23.10 ± 0.22 | 24.57 ± 0.10 | 23.30 ± 0.21 |
| HH3 | 28.51 ± 0.32 | 31.45 ± 0.33 | 29.17 ± 0.66 | 26.41 ± 0.19 | 27.25 ± 0.17 | 26.50 ± 0.30 |

average *Cq* of the 13 genes ranged from 19.26 to 30.36 (Table 5). We found that *EF-1α* and *COX1* had the lowest mean *Cq* values, which represented the highest expression level, in both the total for all stages and separately for each of the five stages, whereas we found that *HH3* had the highest mean *Cq* values in all of the different stages, which indicates it had the lowest expression levels. Interestingly, stages 1 and 2 showed more variation in gene expression compared to that found in the other stages, and that integral *Cq* values decreased gradually from stage 1 to stage 3 and were then constant (Table S1). According to computed SE values, we found that *HH3* (SE = 0.32), *EF-1α* and *NDUFA7* (both SE = 0.34), and *GAPDH* (SE = 0.35) had the least varying transcript abundance when all stages were analyzed together. However, the tested housekeeping genes showed different expression states in each stage under the same condition. The gene with the lowest SE value for each of the five stages is *RL28* (SE = 0.31) and *COX1* (SE = 0.32) in stage 1, *PPIA* (SE = 0.62) in stage 2, *TUBB* and *RS25* (both SE = 0.15) in stage 3, *PPIA* and *TUBB* (both SE = 0.10) in stage 4, and *RS25* (SE = 0.19) in stage 5. These findings illustrate that no single candidate housekeeping gene was expressed at a stable level on the basis of *Cq* value only across five stages from different *R. venosa* larvae samples, and therefore, further statistical analyses were required to identify the best housekeeping genes.

GeNorm-derived *M* values of the candidate housekeeping genes for all stages together and stage-specific are shown in Fig. 3. *COX1* and *RL28* (both *M* = 0.34) have the highest stability in all the stages together. However, we found that *UBE2* and *PPIA* were the most suitable combination of two internal controls for stage 1, *RL5* and *RL28* were optimal for stage 2, *NDUFA7* and *RL28* in stage 3, *RS25* and *PPIA* in stage 4, and *COX1* and *RL5* in stage 5. These results indicate that *RL28* is the most appropriate internal control gene for most of the different stages examined. In addition, Figure 3 shows the ranked stability value calculated by NormFinder of the tested candidate housekeeping genes, and we found that the best housekeeping gene for all combined stages was *RL28* with a stability value of

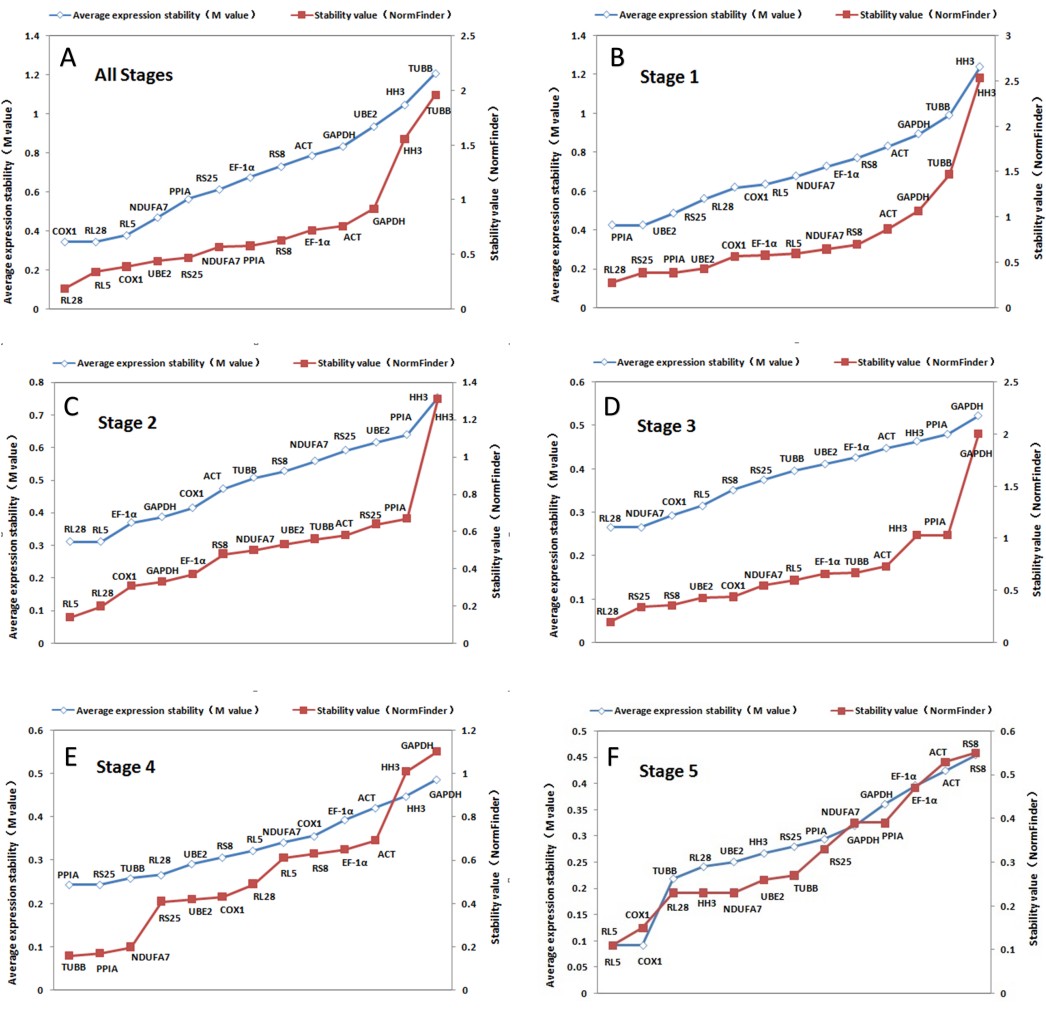

**Figure 3** **Ranking of candidate housekeeping genes in developmental stages.** NormFinder (stability value, filled squares) and GeNorm (average expression stability ($M$ value) of remaining genes, open rhombus) ranking of candidate housekeeping genes in five developmental stages together and separately. A lower value indicates more stable expression.

0.188. It was also the best gene for stages 1 and 3 (0.277 and 0.228, respectively), whereas *RL5* was the best gene for stages 2 and 5 with respective stability values of 0.142 and 0.112, and *TUBB* (0.022) for stage 4. Based on these findings, *RL28* and *RL5* are the appropriate stable internal control genes for most developmental stages.

Based on SD values generated from BestKeeper, we found that various housekeeping genes manifested major differences in expression in each stage (Tables 4 and 6). We identified *HH3* as the best gene for all stages together, whereas the best stage-specific genes were *RL28* for stage 1, *NDUFA7* for stage 2, *TUBB* for stage 3, and *RS25* for stages 4 and 5. However, when ranking by r, we found that *RL28* was the most stably expressed gene when all stages were combined and for stage 3 with *PPIA*, *RL5*, *TUBB*, and *GAPDH* being the ideal genes for stages 1, 2, 4, and 5, respectively.

**Table 6  Ranking of candidate reference genes in order of expression stability calculated by BestKeeper for different developmental stages.** Data shown as Pearson correlation coefficient ($r$) and standard deviation (SD).

| Rank position | All stages | | Stage 1 | | Stage 2 | | Stage 3 | | Stage 4 | | Stage 5 | |
|---|---|---|---|---|---|---|---|---|---|---|---|---|
| | SD | r | SD | r | SD | r | SD | r | SD | r | SD | r |
| 1 | HH3 | RL28 | RL28 | PPIA | NDUFA7 | RL5 | TUBB | RL28 | RS25 | TUBB | RS25 | GAPDH |
| 2 | GAPDH | RL5 | COX1 | RL28 | PPIA | RL28 | RS25 | NDUFA7 | NDUFA7 | COX1 | PPIA | EF-1α |
| 3 | NDUFA7 | RS25 | HH3 | RS25 | HH3 | RS8 | UBE2 | RL5 | TUBB | EF-1α | UBE2 | RL5 |
| 4 | EF-1α | RS8 | RL5 | UBE2 | GAPDH | COX1 | RL28 | COX1 | UBE2 | PPIA | RL28 | RL28 |
| 5 | PPIA | COX1 | EF-1α | RS8 | UBE2 | TUBB | COX1 | RS8 | RS8 | ACT | HH3 | RS25 |
| 6 | COX1 | UBE2 | UBE2 | TUBB | EF-1α | GAPDH | RS8 | RS25 | PPIA | NDUFA7 | TUBB | COX1 |
| 7 | RL5 | NDUFA7 | RS25 | EF-1α | RL28 | EF-1α | NDUFA7 | UBE2 | RL28 | RL28 | COX1 | NDUFA7 |
| 8 | RL28 | ACT | NDUFA7 | NDUFA7 | COX1 | NDUFA7 | HH3 | ACT | RL5 | UBE2 | RL5 | ACT |
| 9 | UBE2 | PPIA | ACT | RL5 | RS25 | ACT | EF-1α | EF-1α | HH3 | RL5 | RS8 | UBE2 |
| 10 | RS25 | EF-1α | PPIA | GAPDH | RL5 | UBE2 | RL5 | PPIA | COX1 | RS25 | NDUFA7 | HH3 |
| 11 | ACT | TUBB | RS8 | COX1 | ACT | RS25 | GAPDH | HH3 | ACT | GAPDH | GAPDH | TUBB |
| 12 | RS8 | GAPDH | GAPDH | ACT | RS8 | PPIA | ACT | TUBB | EF-1α | HH3 | ACT | PPIA |
| 13 | TUBB | HH3 | TUBB | HH3 | TUBB | HH3 | PPIA | GAPDH | GAPDH | RS8 | EF-1α | RS8 |

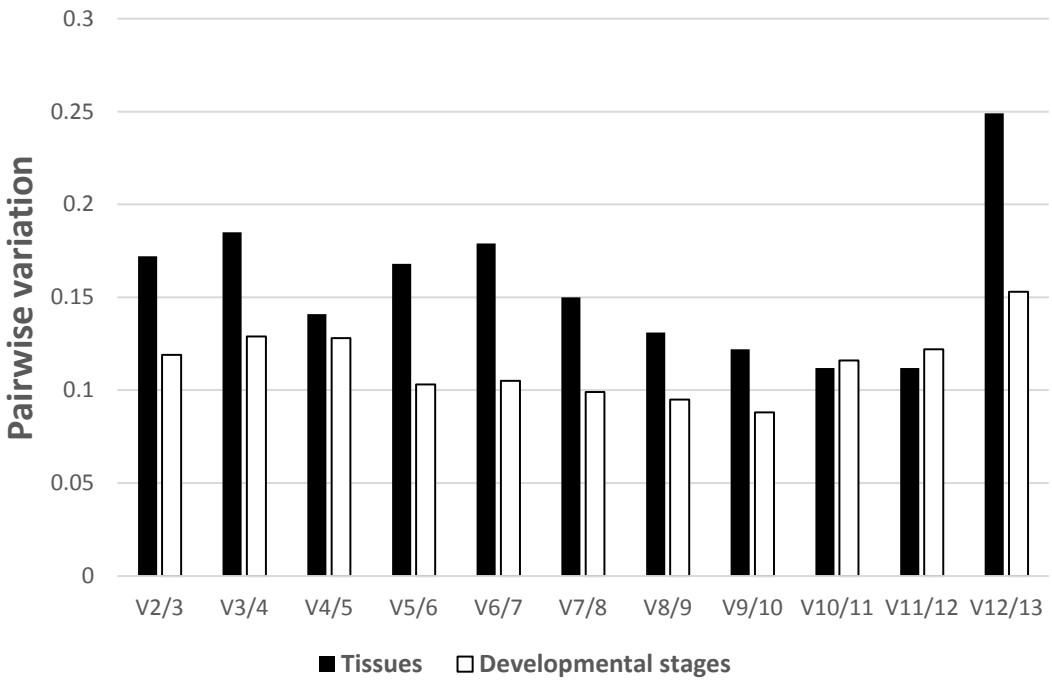

**Figure 4** **Determination of the number of reference genes required for accurate normalization.** Pairwise variation by GeNorm between candidate genes in tissues (black bar) and developmental stages (white bar).

## Determination of the optimal number of internal controls for normalization

In regards to the tissue-specific pairwise variation calculated by GeNorm (Fig. 4), we found that in most tissues (gill, gonad, intestine, Leiblein, kidney, and mantle) the V4/5 value of 0.141 indicated *EF-1α* and *RL5* are insufficient for normalization and that *RL28* and *NDUFA7* should be included; however, the V2/3 value was under 0.15, which suggests two internal control genes were sufficient. In liver, the V2/3 value (0.148) was comparable to the cut-off value although the V3/4 value (0.092) was sufficiently low, which indicates the inclusion of a third reference gene is needed to improve stability of normalization (Fig. S2). We found that none of the pairwise variations determined before V10/11 was less than 0.15 in hemocyte, and therefore, over 10 genes are suitable as reference genes based on the determined conditions (Fig. S4). In terms of pairwise variation for all developmental stages (Fig. 4), we found that the V2/3 value was below threshold (0.15), which suggests *RL28* and *COX1* (on the basis of the *M* value) are sufficient for normalization, and that inclusion of an additional reference gene is not required in most stages with the exception of stage 1. We found in stage 1 that the V2/3 value exceeds threshold, whereas V3/4 is below threshold, which indicates the necessity of adding a third reference gene (*RS25* based on *M* value) to improve the robustness of normalization (Fig. S3).

## DISCUSSION

The commercial importance and ecological role of *R. venosa* have driven increased molecular research towards investigating the morphology and biology of this organism, which may commonly use qRT-PCR as a tool to study gene expression (*Lu et al., 2008*; *Samadi & Steiner, 2009*). It is imperative to study the expression patterns of specific genes in different larval developmental stages and adult tissues in *R. venosa*, and qRT-PCR is a demonstrably powerful tool to analyze such gene expression. Nevertheless, internal controls are critical to obtain reliable normalization of gene expression, and in turn, robust results from qRT-PCR analysis. The ideal internal control gene is characterized by stable expression across different environmental conditions and physiological states, such as different developmental stages and tissue types. However, according to findings from many related studies, housekeeping genes have variable expression changing under different experimental conditions. Thus, no gene has stable expression under all experimental conditions. As a consequence, using a single control gene in all experimental conditions could influence the accuracy of normalization of gene expression results (*Jain et al., 2006*). Therefore, evaluating the level of candidate housekeeping genes is a vital preliminary effort to reliably quantifying target genes by qRT-PCR (*Bustin, 2009*; *Dheda et al., 2005*). Prior to our study, there has been no published finding of research that has examined the suitability of potential housekeeping genes for gene expression analysis in *R. venosa*. We selected 13 genes as candidate internal controls from high-throughput RNA-seq data of *R. venosa* larvae, which showed high and stable levels of expression, and most of which were currently being used as internal control genes in studies of Mollusca. The expression levels of these 13 genes were monitored by qRT-PCR in eight different tissues and twelve developmental larval stages.

We found from comprehensive analysis using GeNorm, NormFinder, and BestKeeper in eight different tissues analyzed together that *EF-1α* was the most stable internal control gene followed by *RL28*. According to the pairwise variation calculated by GeNorm, two additional genes should be added for normalization. Based on the rank determined by NormFinder and BestKeeper, *RS8* and *RS25* are the most appropriate internal controls; however, *RL5* and *NDUFA7* are determined to be the most appropriate internal controls based on GeNorm. Considering these findings, we conclude that *EF-1α*, *RL28*, *RL5*, and *RS8* are the most stable gene combination for *R. venosa* tissues. *EF-1α* belongs to the G-protein family, which has a significant influence in protein translation (*Browne & Proud, 2002*; *Ejiri, 2002*). *EF-1α* was the most stable reference gene in studies of disk abalone exposed to tributyltinchloride and 17β-estradiol (*Wan et al., 2011*), in Atlantic salmon (*Nilsen et al., 2005*), during larval development in flatfish (*Infante et al., 2008*), in hemocytes of flat oyster *Ostrea edulis* (*Morga et al., 2010*), and in different stages of gametogenesis in the mussel, *Mytilus edulis* (*Cuberoleon et al., 2012*). RL5 and RL28 belong to the large subunit ribosomal protein family, while RS8 and RS25 belong to the small subunit family, and they are present in all cell types involved in biogenesis of new proteins. Studies found that *RL5* was the most stable gene in all tissues in red abalone (*López-Landavery et al., 2014*) and in disk abalone following exposure to tributyltinchloride and 17β-estradiol (*Wan et al., 2011*). The other genes relating to ribosomal protein biosynthesis are commonly

considered as housekeeping genes in many other organisms, including animals, plants, and algae (*Barsalobres-Cavallari et al., 2009*; *Hsiao et al., 2001*; *Liu et al., 2012*). *RL7* and *RS18* maintain considerable expression stability in both OsHV-1 infected and uninfected pacific oyster larvae (*Du et al., 2013*). *RS18* was the most stable gene in *Mya arenaria* after *Vibro splendidus* 7SHRW challenge (*Mateo et al., 2010*) and in the intestine of the sea cucumber, *Apostichopus japonicas*, during normal growth and aestivation (*Zhao et al., 2014*).

In addition, using GeNorm, NormFinder, and BestKeeper we identified tissue-specific expression levels of 13 candidate internal controls in eight different tissues to determine the most stable gene for each tissue type. We found that *EF-1α* and *NDUFA7* are the best combination for normalization in gonads. For gills, *EF-1α* and *COX1* are the best combination. COX1, which is encoded from an approximately 650 bp fragment of the mitochondrial gene, is used to identify animals and plants (*Evans, Wortley & Mann, 2007*), but has been rarely used as an internal control of gene expression (*Kaweesi et al., 2014*). However, in this study, we found it demonstrates relatively stable expression and in fact can be the secondary internal control for *R. venosa* tissues and developmental stages. In addition, we identified *COX1* and *RL28* for intestine, *EF-1α* and *RL5* for kidney, and *EF-1α* and *RL28* for Leiblein and mantle. For livers, we determined from our analysis that at least three internal controls are required for reliable normalization, which are *EF-1α* , *RL5*, and *NDUFA7*. In regards to hemocyte, we determined based on pairwise variation that more than 10 genes should be used as reference genes, and therefore, we proposed using as many internal controls available including *GAPDH*, *PPIA*, and *RL28*. Because of our comprehensive assessment, it is evident that *EF-1α* is a good housekeeping gene as an internal control in most *R. venosa* tissues and for a particular tissue, we recommend including additional corresponding housekeeping genes to improve stability.

Comprehensive analysis using GeNorm, NormFinder, and BestKeeper combining different developmental stages in *R. venosa* identified *RL28* and *RL5* as the most stable gene combination for normalization, a finding that is supported from an assessment of the pairwise variation. In contrast, three genes are required for normalization in stage 1, namely, *PPIA*, *RS25*, and *RL28*. *PPIA* has been used as an internal control on account of its stable expression in milk somatic cells under healthy and disease status (*Jarczak, Kaba & Bagnicka, 2014*). Furthermore, *PPIA* was also found to be a stable reference gene in the mammary gland of goats, murines, and bovines (*Bonnet et al., 2013*; *Boutinaud et al., 2004*; *Robinson, Sutherland & Sutherland, 2007*). With the exception of stage 1, the other four stages require two genes for normalization, specifically, *RL5* and *RL28* for stage 2 and stage 5, *RL28* and *NDUFA7* for stage 3, and *PPIA* and *TUBB* for stage 4. Both α-tubulin (TUBA) and TUBB belong to the tubulin family of proteins, and are considered suitable internal controls in research because of their high expression stability. *TUBB* was found to be the most stable gene in different developmental stages of *Hippoglossus hippoglossus* (*Fernandes et al., 2008*) and in goat follicles (*Costa et al., 2012*). In addition, two isoforms of *TUBB* (*TUB1* and *TUB5*) are used as control genes in *Striga hermonthica* (*Fernández-Aparicio et al., 2013*). Interestingly, in the current study, we found that *TUBB* showed significantly different expression stability in *R. venosa*. *TUBB* was the least stable gene when all tissues and stages were assessed together as well as in gonads and gills, whereas in contrast it was the most stable

gene in stage 4. This difference illustrates the importance of studies such as ours to identify and evaluate species-specific housekeeping genes for qRT-PCR as differences in expression of the same gene exists between different species. Based on our findings, we determined that *RL28* was an appropriate housekeeping gene for use as an internal control for qRT-PCR during most developmental stages of *R. venosa* larvae. Furthermore, we propose that *RL5* and other specific genes can be included in normalization as needed for the corresponding developmental stage.

*GAPDH*, *ACT*, *UBE2*, and *HH3* are housekeeping genes that are considered as internal controls for qRT-PCR in both plants and animals. However, these genes showed unsuitability as internal controls in *R. venosa* adult tissues and larvae development in our study. *GAPDH* is expressed stably under different tissue types in *Crassostrea gigas* (*Dheilly et al., 2011*) and hence it was selected as internal control for qRT-PCR analysis of *R. venosa* in previous study (*Song et al., 2016*). According to present results, *GAPDH* is not recommended as an internal control for future studies involved with *R. venosa* adult tissues and larvae development. As to *HH3*, in this study, we identified it according to GeNorm and NormFinder as the least stable gene for most tissues (intestine, kidney, Leiblein, liver, and mantle) and for stages 1 and 2. These results demonstrate that *HH3* is unsuitable to be used as an internal control in *R. venosa*.

So far, there is no ideal internal control that can be fully used for various types of samples with constant expression. Usually, housekeeping genes have variable expression changing under different conditions. As a consequence, using a single control gene in all experimental conditions could influence the accuracy of normalization of gene expression results (*Jain et al., 2006*). To eliminate bias and variation of these, *Vandesompele et al. (2002)* suggested to use three control genes to correct during normalization at least. Therefore, in order to investigate the expression of target gene, researchers should take all conditions into account, including types of sample, species, gene family and experiment conditions, and choose suitable and stable gene to normalize to acquire reliable data.

## CONCLUSIONS

We identified and evaluated expression stability of 13 housekeeping genes for qRT-PCR normalization in *R. venosa* tissues and larvae developmental stages. In our assessment of tissue-specific genes, *EF-1α* was the most stable internal control gene in most tissue samples tested with *RL5* and *RL28* as suitable secondary choices. We found that *RL28* was the most stable gene when evaluating all measured developmental stages, and *COX1* and *RL5* were appropriate secondary choices. To our knowledge, this study was the first to investigate and identify optimal housekeeping genes for relative quantification of qRT-PCR in *R. venosa*. The results of this study provide not only references to estimate gene expression levels during *R. venosa* larvae developmental stages, but also enables future research efforts to measure readily and robustly tissue-specific mRNA abundance in *R. venosa*.

**List of abbreviations**

| | |
|---|---|
| **qRT-PCR** | quantitative real-time PCR |
| **EF-1$\alpha$** | elongation factor-1$\alpha$ |
| **ACT** | $\alpha$-actin |
| **COX1** | cytochrome c oxidase subunit 1 |
| **NDUFA7** | nicotinamide adenine dinucleotide dehydrogenase [ubiquinone] 1$\alpha$ subcomplex subunit 7 |
| **RL5** | 60S ribosomal protein L5 |
| **RL28** | 60S ribosomal protein L28 |
| **GAPDH** | glyceraldehyde 3-phosphate dehydrogenase |
| **TUBB** | $\beta$-tubulin |
| **RS25** | 40S ribosomal protein S25 |
| **RS8** | 40S ribosomal protein S8 |
| **UBE2** | ubiquitin-conjugating enzyme E2 |
| **HH3** | histone H3 |
| **PPIA** | peptidyl-prolyl cis-trans isomerase A |
| *Cq* **values** | quantification cycles |
| **Leiblein** | Leiblein's gland |

### Funding

The research was supported by the Project supported by the National Natural Science Foundation of China (Grant No. 31572636), the NSFC-Shandong Joint Fund for Marine Science Research Centers (Grant No. U1606404), the Strategic Priority Research Program of the Chinese Academy of Sciences (XDA11020703), the Agricultural Major Application Technology Innovation Project of Shandong Province. The funders had no role in study design, data collection and analysis, decision to publish, or preparation of the manuscript.

### Grant Disclosures

The following grant information was disclosed by the authors:
National Natural Science Foundation of China: 31572636.
NSFC-Shandong Joint Fund for Marine Science Research Centers: U1606404.
Strategic Priority Research Program of the Chinese Academy of Sciences: XDA11020703.
Agricultural Major Application Technology Innovation Project of Shandong Province.

### Competing Interests

The authors declare there are no competing interests.

### Author Contributions

- Hao Song conceived and designed the experiments, performed the experiments, analyzed the data, wrote the paper, prepared figures and/or tables, reviewed drafts of the paper.
- Xin Dang performed the experiments, analyzed the data, wrote the paper, prepared figures and/or tables, reviewed drafts of the paper.

- Yuan-qiu He performed the experiments, wrote the paper, reviewed drafts of the paper.
- Tao Zhang and Hai-yan Wang conceived and designed the experiments, contributed reagents/materials/analysis tools, reviewed drafts of the paper.

## Field Study Permissions

The following information was supplied relating to field study approvals (i.e., approving body and any reference numbers):

The collecting of the egg capsules and adults of *R. venosa* in Laizhou Bay was permitted by Ren-tao Kan, manager of Blue Ocean Co. Ltd.

## Data Availability

The raw data has been supplied as a Supplementary File.

## Supplemental Information

Supplemental information for this article can be found online at http://dx.doi.org/10.7717/peerj.3398#supplemental-information.

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
