# Peer review of "Selection of housekeeping genes as internal controls for quantitative RT-PCR analysis of the veined rapa whelk (Rapana venosa)"

_PeerJ, doi:10.7717/peerj.3398_

## Round 0.1 · original submission · Minor Revisions

· Academic Editor

Minor Revisions

Please address each of the reviewers' comments, with one exception: you are under no obligation to include as a reference the Bangaru et al. paper recommended by Dr Bangaru.

In addition, I recommend:

1. changing the phrase "developmental larvae" to "developmental larval stages" throughout.

2. stating clearly how many larvae of each developmental stage were used to create the 5 pooled stages (lines 235-239). This should be cross-referenced with the answer to Dr Kuchipudi's first point regarding total RNA/cDNA input, since the phrase "five biological replicates" (line 102) indicates that five individual larvae at each stage were pooled. If this is so, then stage 1 would include 15 individuals (five each of L, M and N) whereas stage 5 would include only 5 juveniles. Such differences in RNA input and source could significantly bias cDNA production and input into PCR.

Reviewer 1 ·

Basic reporting

no comment

Experimental design

no comment

Validity of the findings

no comment

Additional comments

This manuscript presents a first study to select stable housekeeping genes as internal controls for qRT-PCR in the veined rapa whelk Rapana venosa. The experiments appear to be performed according to high research standards and presentation of results is clear and easy-to-follow. The authors also provided necessary supplementary data to aid readers interpreting the study’s results. In particular, the discussion is highly relevant to the research theme and suggests logical and comparative explanations for the reliable internal control selecting in the context of different developmental stages and tissues. Overall, the work ultimately contribute to advancement in studying gene expression in R. venosa and its related species. It could be accepted for publication in Peer J after minor revision.

The manuscript is well written and the use of English is excellent. Here are minor points which need clarification or correction:

Line 19 there is no need to use parenthesis to include the name Rapana venosa
Line 39-40 “for gill”, “for liver”, “for hemocyte” should not be italic
Line 53, delete available
Line 61, relative qRT-PCR assay
Line 87, delete first
Line 101-103 “We selected five biological replicates from each larvae stage and all tested tissues (gill, hemocyte, intestine, Leiblein’s gland, liver, kidney, mantle, and gonad) were aseptically dissected from five adult specimens.” Please clarify which developmental stages you sampled. As noticed that you had described this in result part (Line 235-239), I recommend to move this into M&M part in line 102.
Line 104, Hemolymph was extracted from the pericardial cavity. Hemocytes were obtained by centrifugation at which condition ?
Line 114-115 “According to RNA-seq transcriptome data of developmental samples, which were derived from stages C to Y and performed in triplicate for each stage (Song et al. 2016)” What does C and Y means?
Line 195, 196, the exception of mantle, in various
Line 278-286 I think the author is stating pairwise variation calculated by GeNorm in terms of single tissues, which is not shown in Figure 4, thus a supplementary figure is required.
Line 296, which may not research which may
Line 320~321, the most appropriate internal controls

·

Basic reporting

No comment

Experimental design

No comment

Validity of the findings

No comment

Additional comments

The manuscript with the title “Selection of housekeeping genes as internal control for quantitative RT-PCR analysis of the veined rapa whelk (Rapana venosa)” is an elaborate and thorough study towards identifying the stable reference genes in the adult tissues and the developmental stages of Rapana venosa.
The authors have selected 13 different known housekeeping genes and identified those genes that show stable expression levels using three different well known statistical analysis programs.
This study contributes to improvement and accuracy of the gene expression studies measured by the qRT PCR analysis that will be carried out by all the groups working with this model system.
Corrections
1. According to the data in Table 5 (Line no 652), the following changes have to be done
• Line no 274-“TUBB for stages 3 and 4, and RS25 for stage 5” has to be corrected to “TUBB for stage 3 and RS25 for stages 4 and 5”.
• Similarly,Line 276- “with PPIA, RL5, RS25, and GAPDH being the ideal genes for stages 1, 2, 4, and 5 respectively.” has to be corrected to “with PPIA, RL5, TUBB, and GAPDH being the ideal genes for stages 1, 2, 4, and 5 respectively.”

2. A similar study that has been carried out in dorsal root Ganglia neurons after peripheral nerve injury in the rat model system can be mentioned in the introduction
• Bangaru, M.L.Y., Park, F., Hudmon, A., McCallum, J.B. and Hogan, Q.H., 2012. Quantification of gene expression after painful nerve injury: validation of optimal reference genes. Journal of Molecular Neuroscience, 46(3), pp.497-504.

3. Line 409- “There is no an ideal control can be used” has to be changed to “There is no ideal internal control that can be used”

·

Basic reporting

The study by Song et al investigated expression stability of 13 candidate gens to identify a suitable reference gene for gene expression analysis in veined rapa whelk (Rapana venosa),an important commercial shellfish in China. The manuscript is well written with adequate background literature and providing sufficient details about the methods.

Experimental design

The study is well designed with sufficient sample size and used relevant tools for analyzing the expression stability of candidate reference geens

Validity of the findings

The data presented is sound and the conclusions drawn are supported by the data

Additional comments

The study is well designed and the data is presented well in the manuscript. However, I have the following comments to improve the clarity of the manuscript.

Methods:
1. Total RNA extraction and cDNA conversion-(Line 110)- It is not clear how much total RNA was used for cDNA conversion. the authors should provide this detail and also confirm whether same amount of total RNA was used for cDNA conversion across all the samples.

2. Selection of candidate internal controls-(Line 116)- The authors mentioned that they identified genes with similar expression patterns from the RNA-seq data. However, it is not clear what were the metrics, for example is it based on RPKM values or others. This needs to be more specific.

3. Primer design and qRT-PCR-(Line 130)- It is mentioned that the gel pictures were analyzed top confirm specificity of the PCR product. I am not sure if the authors actually mean that they confirmed the product by the predicted size. The best practice is to sequence the product to confirm the specificity of the reaction. The authors should clearly state that their assessment was based on the product size.

Results:
1. Selection of house keeping genes: (Line 173); The authors should clarify that the whole process of selection of the 13 genes (sub clusters etc) was done using a previously published RNA-seq data (Song et al 2016). This information should probably be under methods section rather than in Results.
2. (Line 185)- Same comment as Point 3 above, the authors should add that the confirmation of PCR product was based on the expected size.

Discussion:
The discussion section is too long and a lot of not so relevant literature has been discussed. They did not discuss mush about the relevant literature pertaining to veined rapa whelk.
The Discussion section needs to be shortened focusing on discussing more relevant literature. For example, The authors highlighted that gene expression analysis of veined rapa whelk is becoming popular and no previous study looked at expression stability of house keeping gees. Based on the results of this study, the authors should address questions like, which genes were used as house keeping for data normalization in published studies with veined rapa whelk and how the results of this study would inform any future studies

Minor points:
Introduction:
1. (Line 75) Its is not clear what the authors mean by "housekeeping genes that can be used to verify the validity of the qRT-PCR results". Hose keeping genes are essential for data normalization and hence analysis of Q RT-PCR data but not used to verify the validity of qRT-PCr data.
2. (Line 85) - "Therefore, because",,,,,the sentence is clumsy and needs rewriting

---

## Round 0.2 · accepted · Accept

· Academic Editor

Accept

I have evaluated your revision and rebuttal and am happy to Accept this manuscript. Thank you for choosing Open Access, and PeerJ in particular!